

# Beyond predefined ligand libraries: a genetic algorithm approach for *de novo* discovery of catalysts for the Suzuki coupling reactions

Julius Seumer and Jan H. Jensen

Department of Chemistry, University of Copenhagen, Copenhagen, Denmark

## ABSTRACT

This study introduces a novel approach for the *de novo* design of transition metal catalysts, leveraging the power of genetic algorithms and density functional theory calculations. By focusing on the Suzuki reaction, known for its significance in forming carbon-carbon bonds, we demonstrate the effectiveness of fragment-based and graph-based genetic algorithms in identifying novel ligands for palladium-based catalytic systems. Our research highlights the capability of these algorithms to generate ligands with desired thermodynamic properties, moving beyond the restriction of enumerated chemical libraries. Limitations in the applicability of machine learning models are overcome by calculating thermodynamic properties from first principle. The inclusion of synthetic accessibility scores further refines the search, steering it towards more practically feasible ligands. Through the examination of both palladium and alternative transition metal catalysts like copper and silver, our findings reveal the algorithms' ability to uncover unique catalyst structures within the target energy range, offering insights into the electronic and steric effects necessary for effective catalysis. This work not only proves the potential of genetic algorithms in the cost-effective and scalable discovery of new catalysts but also sets the stage for future exploration beyond predefined chemical spaces, enhancing the toolkit available for catalyst design.

## INTRODUCTION

Catalysis plays a crucial role in synthetic chemistry and is fundamentally dependent on the formulation of catalysts that are both efficient and selective. A reaction of critical importance in this domain is the Suzuki coupling because of its ability to synthesize carbon–carbon bonds *via* the coupling of organohalides with boronic acids (*Miyaura, Yamada & Suzuki, 1979*; *Miyaura & Suzuki, 1995*). The performance and specificity of this reaction are largely influenced by the ligand selection in palladium (Pd)-based catalytic systems (*Suzuki, 1991*).

The traditional approaches to ligand discovery have been characterized by intensive experimental screening processes that are both time-intensive and demand significant

Corresponding author
Jan H. Jensen, jhjensen@chem.ku.dk

resources (*Porte, Reibenspies & Burgess, 1998*; *Sigman, Vachal & Jacobsen, 2000*; *Renom-Carrasco & Lefort, 2018*). Computational methodologies for identifying efficient catalysts have evolved, predominantly involving the screening of extensive enumerated libraries, typically ranging from $10^{3-5}$ catalysts (*Meyer et al., 2018*). These virtual screenings are executed through (semi-empirical) quantum mechanics (QM) calculations which limits the size of the screen library due to the computational cost (*Fu et al., 2014*; *Rosales et al., 2018*). In response to this challenge, researchers have turned to machine learning (ML) models, trained on pre-existing data, to screen more extensive libraries, which may include up to $10^6$ catalysts or more (*Nandy et al., 2022*; *Nandy et al., 2021*; *Zahrt et al., 2019*). While ML models have proven valuable within their trained domains, their efficacy often diminishes when applied to scenarios outside their original scope, particularly in the context of true *de novo* generation. Another approach to limit computational cost is to employ search algorithms aimed at efficient navigation of chemical space, such as genetic algorithms (GAs). However, a notable limitation of these methods is their tendency to restrict searches to predefined chemical spaces, often encompassing around $10^{5-7}$ catalysts (*Chu et al., 2012*; *Foscato, Venkatraman & Jensen, 2019*; *Laplaza, Gallarati & Corminboeuf, 2022*; *Kneiding, Nova & Balcells, 2023*; *Gallarati et al., 2024*; *Foscato et al., 2024*).

Recently, our research has introduced the *de novo* design of a highly efficient organic homogeneous catalyst, specifically devised for the Morita–Baylis–Hillman reaction (*Seumer et al., 2023*). Utilizing a genetic algorithm, we explored the unrestricted chemical space of tertiary amines, signifying a shift from traditional screening methods and restricted chemical spaces.

Following that, *Strandgaard et al. (2023)* have shown the computational *de novo* design of fragments of ligands for the Schrock catalysts which extends the unrestricted genetic algorithm search to parts of inorganic homogeneous catalysts.

In this study we expands this concept to demonstrate the *de novo* design of whole ligands for transition metal catalysts at the example of the Suzuki reaction. While ML models trained on pre-existing data have proven effective in high-throughput screening, their use for unrestricted *de novo* design is challenging since their predictive performance on truly out of domain samples deteriorates. Another approach, relying on the correlation between geometric descriptors such as bond-lengths, cone angles or sterimol parameters and the thermodynamics of the catalytic sytem has been shown to be effective, yet it appears challenging when moving beyond only one binding motif (*Chu et al., 2012*; *Brethomé, Fletcher & Paton, 2019*; *Foscato et al., 2024*). Therefore, we evaluate the performance of a catalyst by calculation of density functional theory (DFT) level thermodynamic descriptors, instead of using ML models or (semi-empirical) QM calculated geometric descriptors. This distinctive approach expands the domain of our *de novo* design, allowing for the exploration of chemical space beyond the confines of ML model training or QSAR correlations.

## COMPUTATIONAL METHODOLOGY

We use a genetic algorithm (GA) to discover promising catalyst candidates for the Suzuki reaction. In the GA, the gene is represented as a list of two molecular fragments which are

the ligands of the Suzuki catalyst. Two different GAs are used which differ only in their reproduction rules.

In the fragment-based GA (FBGA), a crossover operation means that one ligand is exchanged for another ligand from another catalyst. The mutation operation exchanges one of the two ligands with another selected randomly from a pre-defined list of ligands.

The graph-based genetic algorithm (GBGA) utilizes the crossover and mutation operations as implemented by *Jensen (2019)*. During a crossover operation, the graph of one of the ligands is cut at random points and recombined with a fragment of a molecular graph from another individual. The mutation operations act directly on the molecular graph, adding, changing or removing atom(-type) or connectivity. Newly generated ligands are considered valid if they contain at least one molecular pattern that is considered to be a potential coordination site for the ligand. These patterns are phosphines, amines, carbenes and carbonyls. When more than one potential coordination site is detected, all possible constitutional conformers of the ligand attached to a Pd-containing reference catalyst are generated and 25 conformers of each are embedded using ETKDG and optimized at the GFN2-xTB level of theory (*Bannwarth, Ehlert & Grimme, 2019*). The coordination site with the highest binding energy is then chosen as the coordination site for that ligand.

All GAs are run with a population size of 25 for up to 50 generations. The mutation rate is set to 50 % which means that with a 50:50 chance either a crossover or mutation operation is chosen for the reproduction. Ligands with as many as 30 heavy atoms and/or five rotatable bonds are allowed. The starting population is created from combinations of ligands from a pre-defined list containing 91 different amines, phosphines, N-heterocyclic carbenes, pyridines and CO taken from *Meyer et al. (2018)*. A randomly picked subset of 12 ligands is shown in Fig. S1.

For both GAs, the fitness of each individual depends on the difference in electronic energy of intermediate **2** and **1**, see Fig. 1. As *Meyer et al. (2018)* established using linear-energy scaling relationships following an approach by *Busch, Wodrich & Corminboeuf (2015)*; *Wodrich, Busch & Corminboeuf (2016)*, the optimal difference in electronic energy ($\Delta E$) between the two intermediates is in the range of $-32.1$ and $-23.0$ kcal mol$^{-1}$ at the B3LYP-D3BJ/def2-TZVP//B3LYP-D3BJ/3-21 level (*Becke, 1993*; *Lee, Yang & Parr, 1988*; *Vosko, Wilk & Nusair, 1980*; *Stephens et al., 1994*; *Becke & Johnson, 2005*; *Grimme, Ehrlich & Goerigk, 2011*; *Binkley, Pople & Hehre, 1980*; *Weigend & Ahlrichs, 2005*). We calculate $\Delta E$ at the B3LYP-D3BJ/def2-TZVP//B3LYP-D3BJ/3-21 level and convert it to a score between 0 and 1 using a Gaussian function centered around $-27.55$ kcal mol$^{-1}$ ($\mu_1$) and a standard deviation ($\sigma_1$) of 6.00 kcal mol$^{-1}$, chosen empirically based on the distribution of $\Delta E$ values of the starting population, see Eq. (1). Therefore, the closer the calculated $\Delta E$ is to the target value of $-27.55$ kcal mol$^{-1}$ the closer the score is to 1.0.

To calculate $\Delta E$ for a molecule, 10 conformers of intermediate **2** are embedded using ETKDG as implemented in a slightly modified version of RDKit based on 2023.03.2, see Subsection S1, and an RMSD pruning threshold of 0.25 Å is applied (*Riniker & Landrum, 2015*; *Landrum et al., 2023*). The retained conformers are optimized at the GFN2-xTB level of theory (*Bannwarth, Ehlert & Grimme, 2019*). The lowest energy conformer is further optimized at the B3LYP-D3BJ/3-21 level and its single point energy is calculated

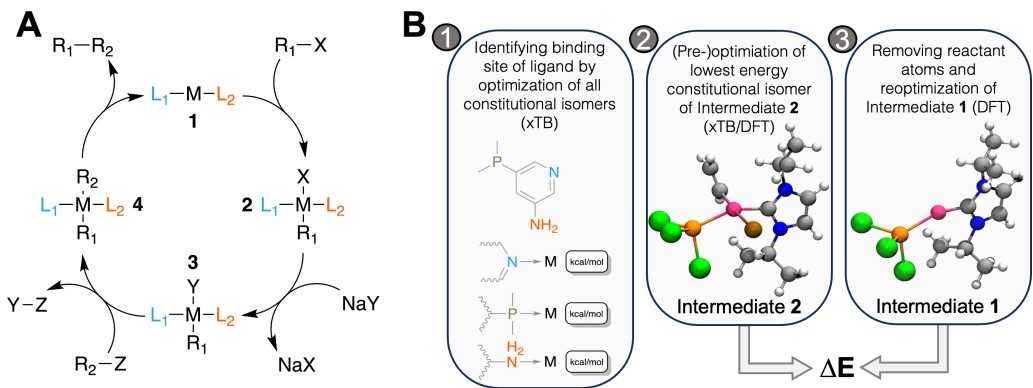

**Figure 1** (A) Catalytic cycle of the Suzuki reaction with key intermediates 1 and 2, (B) workflow to calculate ΔE for a catalyst: 1. for ligands with multiple potential binding sites, all constitutional isomers are generated, one conformer is generated for each and its structure optimized with GFN2-xTB. For the lowest energy isomer, nine additional conformers are generated and optimized with GFN2-xTB. The lowest energy structure is further optimized at the DFT level in step 2. The atoms corresponding to the reactant (**R₁-X**) are removed and the structure is optimized in step 3. ΔE is then obtained as the difference in electronic energy at the DFT level. The coloring scheme for the atoms in the rendering is the following: carbon, grey; hydrogen, white; nitrogen, blue; phosphorus, orange; chlorine, green; bromine, brown; palladium, pink.

as the B3LYP-D3BJ/def2-TZVP level using ORCA 5.0.4 (*Neese, 2022*). The fragments **R₁-X** (HC=CH2) and **X** (Br) are removed from the optimized structure to obtain a structure of intermediate **1** which undergoes optimization and single point calculation at the B3LYP-D3BJ/def2-TZVP//B3LYP-D3BJ/3-21 level of theory. ΔE is then obtained as the difference in electronic energy between intermediate **2** and intermediate **1** and **R₁-X**.

In the FBGA, the final score of individual $i$ is equal to the first factor in Eq. (1) which only depends on $\Delta E_i$. In the GBGA, this score is multiplied with a modified synthetic accessibility (SA) score, which also ranges from 0.0 to 1.0 as shown in Eq. (1). The original SA score for a ligand is calculated based on a fragment-frequency contribution and a structure based penalty. The fragment-frequency contribution evaluates how often radial fragments of the input ligand molecule are present in a subset of PubChem, where common fragments give a positive contribution and rare fragments give a negative contribution to the estimated synthesizability. This estimate is further augmented with a structure-based penalty which penalizes complex motifs such as macro-cycles, stereocenters, bridgehead- or spiro-atoms, *etc.*

Here, the modified SA score is the mean of the two SA scores of the ligands which are calculated as described by *Ertl & Schuffenhauer (2009)* and further modified using a modified Gaussian function as proposed by Brown et al. with the parameters $\mu_2 = 2.230044$ and $\sigma_2 = 0.6526308$ as used by Gao and Coley (*Brown et al., 2019*; *Gao & Coley, 2020*). Therefore, the final score in both GAs ranges from 0.0 to 1.0.

$$\text{Score}_i = \exp\left(\frac{-(\Delta E_i - \mu_1)^2}{2\sigma_1^2}\right) \cdot \exp\left(\frac{-\left(\max\left(\bar{SA}_i, \mu_2\right) - \mu_2\right)^2}{2\sigma_2^2}\right). \quad (1)$$

We choose to calculate the overall score as the product of the two normalised objectives, the energy-dependent term and the synthesizability penalty so that an optimal solution can only be found when both objectives are satisfied since we are not interested in high synthesizable molecules that do not perform well as catalysts or molecules that are calculated to perform well as catalysts but are not synthesizable or show uncommon structural motifs. This is achieved by using the product as shown in Eq. (1), compared to, for example, a sum of the two terms.

Based on the rank $r_i$ of each individual in each population ($N$ = population size), a normalized fitness value is calculated using Eq. (2) from *Baker (1985)* with a selection pressure (SP) of 1.5, chosen empirically to balance exploration and exploitation. Individuals are selected for reproduction with a frequency proportional to their normalized fitness value.

$$p_i = \frac{1}{N}\left(2 - \mathrm{SP} + 2\cdot(\mathrm{SP}-1)\cdot\frac{r_i - 1}{N-1}\right). \tag{2}$$

Therefore, a lower selection pressure leads to higher probability of reproduction for individuals with lower scores and a wider exploration of chemical space, whereas a higher selection pressure results in the exploitation of molecular structures that yield high scores resulting in structurally less diverse molecules.

## RESULTS AND DISCUSSION

### Fragment-based GA

Firstly, we assess the ability of a fragment-based GA to locate catalysts based on Pd within the defined $\Delta$E range. The starting population is comprised of 25 molecules that all have a calculated $\Delta$E below $-42.00$ kcal mol$^{-1}$ ($\Delta$E far below target value). Subsequently, the GA explores if combinations of the present ligands can yield molecules with $\Delta$E values closer to the target of $-27.55$ kcal mol$^{-1}$ and adds new ligands from the full list of available ligands *via* mutation operations (*Meyer et al., 2018*). The evolution of the best-performing molecule over 10 generations is shown in Fig. 2. The score of the best-performing individual quickly increases from 0.04 to 0.99 within two generations. Closer inspection of the GA run reveals that the molecule is created by two successive mutation operations as shown in Fig. 3. After four generations the best-performing molecule has a calculated energy difference of $-27.33$ kcal mol$^{-1}$ which yields a score of 1.00. Over the next six generations, more molecules are found with a score of 1.00 and after ten generations all 25 molecules have a calculated energy difference in the target range as defined by *Meyer et al. (2018)*. Here, we performed $2 \cdot 25 \cdot 10 = 500$ DFT optimization (two for each catalyst, 25 catalysts for 10 generations) to locate a total of 134 unique catalysts within the target range. *Meyer et al. (2018)* were able to identify 265 unique Pd catalysts that are predicted to have an energy difference in the target range using an ML model to exhaustively screen the same library we search with this GA. Their ML model was trained on a total of 7054 molecules out of which 2595 contained Pd. If they restricted themselves to only Pd-containing catalysts and were able to obtain a model performing similarly on Pd-containing catalysts, they would have been able to find 265 catalysts while doing $2 \cdot 2595 = 5190$ DFT optimization (two for

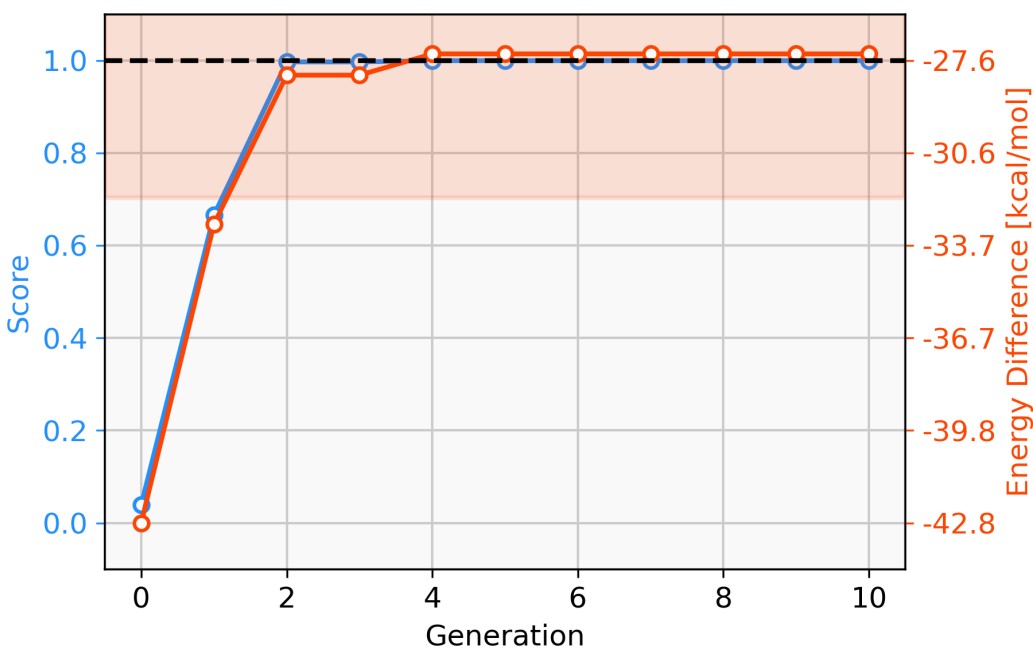

**Figure 2** Evolution of the score (blue) and energy difference (red) of the best-performing individual of an FBGA run over 10 generations. The target range of the energy difference is shown as a red-shaded area.

each molecule). This highlights how GAs can be used in connection with DFT for scoring to search a large pre-defined library of ligands with comparably little computational cost. The catalysts presented here are heteroleptic with two monodentale ligands, following previous studies by *Meyer et al. (2018)* and *Schilter et al. (2023)*. Yet, most of the frequently used catalysts for Suzuki reactions are homoleptic. This constraint could be incorporated in the optimization process, which could yield catalysts which are easier to synthesize. However, transition metal based heteroleptic complexes with two monodentate ligands are not uncommon. In the 250k complexes in the Cambridge Structural Database which contain one transition metal atom, 2097 have exactly two monodentate ligands. Out of these complexes, 85% are heteroleptic, which shows that the synthesis of such complexes is not uncommon.

## Graph-based GA

Next, we aim to discover novel ligands for Pd-containing catalysts for the Suzuki reaction and not just recombine pre-defined ligands.

From the same starting population, a GBGA without synthetic accessibility constraint is run for 20 generations. Here, the best-performing individual is created by subsequent crossover and mutation operations on the molecular graph of the ligands.

Since only parts of one ligand are changed in the GBGA instead of the whole ligand as in the FBGA, evolution of molecular structures happens in smaller steps through chemical space. This allows the discovery of novel structures. The score of the best-performing individual increases drastically over the first four generations and the calculated energy difference increases from $-42.8$ kcal mol$^{-1}$ to $-27.1$ kcal mol$^{-1}$, as shown in Fig. 4.

**Figure 3** **Evolution of the best-performing individual of an FBGA run after two generations.**

Over the following 16 generations, the energy difference of the best-performing molecule decreases slightly to $-27.59$ kcal mol$^{-1}$ which corresponds to a score of 1.00. After 20 generations, all molecules in the population have a calculated energy difference within the target range and the four best-performing individuals are shown in Fig. 5. The ligands coordinate to the transition metal *via* phosphine or pyridine derivative sites and possess up to five hetero atoms.

Although no synthetic accessibility constraint was applied, some purchasable ligands such as 2-fluoropyridine were discovered by the GA (*PostEra, 2024b*). On the other hand,

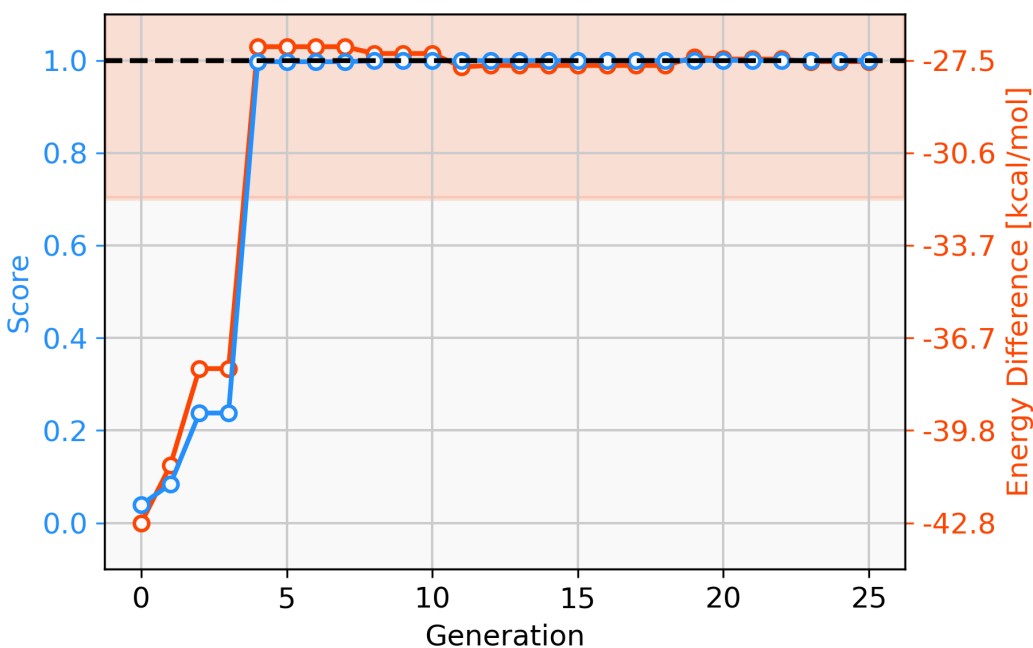

**Figure 4** **Evolution of the score (blue) and energy difference (red) of the best-performing individual of a GBGA run over 25 generations.** The target range of the energy difference is shown as a red-shaded area.

for many of the ligands containing highly fluorinated phosphines, no synthetic route can be found with the retrosynthesis software Manifold (*PostEra, 2024a*).

We predict the binding site of each ligand as the site with the highest binding energy when coordinating as a monodentate ligand. This might not be a reasonable assumption for one of the ligands of the fourth molecule shown in Fig. 5 which might coordinate as an N,N-bidentate ligand. One could perform more automated xTB calculations considering other coordination modes than simple monodentate coordination to verify this in the GA. This way the coordination site and mode could be identified by the highest binding energy across all sites and modes.

Here, we defer this additional consideration to verification and evaluation steps that are necessary after the molecular optimization, along with a more extensive conformational search, location of transition states and calculation of activation barriers, extensive retrosynthetic analysis and calculation of full catalytic cycles.

## Graph-based GA with SA

To address this short-coming, a GBGA is started again from the same starting population with a synthetic accessibility constraint to the score as described in 'Computational Methodology' which steers the search into an area of chemical space that is deemed to be more accessible. Virtually all molecules of the initial population are deemed to be synthetically inaccessible by the modified SA score, as shown in Fig. 6. This is not surprising since it was developed for drug-like molecules. Within ten generations, the GA discovers new ligands that are deemed to have moderate synthetic accessibility as well as an energy difference within the desired target range. Here, a trade-off has to be made between the

-27.59 kcal/mol

-27.51 kcal/mol

-27.45 kcal/mol

-27.42 kcal/mol

**Figure 5  Best-performing individuals of the GBGA after 25 generations.** The calculated energy difference is shown to the right of the structure.

two components of the score, the energy difference and the modified SA score. Although molecules with modified SA scores of 1.00 are found after seven generations, the modified SA score of the best-performing molecule decreases again to 0.6 after ten generations since the energy difference of the molecule is closer to the target value of $-27.55$ kcal mol$^{-1}$ which yields an increase to the overall score of 0.2. On the other hand, after 26 generations, the best-performing molecule has a less favorable energy difference than the best-performing molecule in the previous generation but this is compensated for by a higher modified SA score which yields an overall increased score of 0.1. After 30 generations, molecules with energy differences close to the target and near-perfect synthetic accessibility score are located. Figure S2 shows the cumulative count of catalyst candidates with calculated $\Delta E$ in the target range. From the initial discovery of a catalyst in the target range, the count increases steadily and after 40 generations over 500 unique catalysts candidates are discovered with a total of 1,025 catalysts being evaluated.

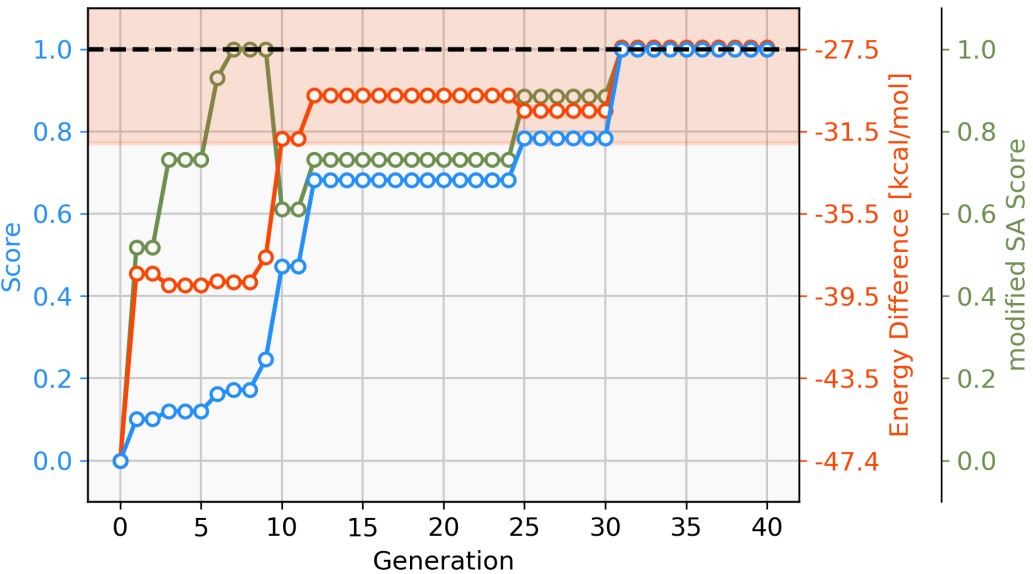

**Figure 6** **Evolution of the score (blue), energy difference (red) and modified SA score (green) of the best-performing individual of a GBGA run over 40 generations.** The target range of the energy difference is shown as a red-shaded area.

The best-performing individuals from the final population are shown in Fig. 7. All molecules possess one ligand coordinating *via* a nitrile group to the transition metal and another one coordinating *via* a carbonyl group or the nitrogen atom of a pyridine derivative. All six unique ligands are purchasable building blocks *via* Sigma-Aldrich and/or Mcule (*PostEra, 2024c*). Figure 8 shows a bar chart of the different coordination sites of the discovered catalysts at different evolutionary steps. In the initial population of the GA, which was selected to have low $\Delta E$ values, N-heterocyclic carbenes (NHCs) are the most common coordination sites followed by pyridine derivatives and phosphines. Neither NHCs nor phosphines are found as binding sites after 20 generations of the GA. This could be partially due to a low modified SA score for phosphines and NHCs. Instead, more pyridine derivatives are found and nitriles are discovered as a favourable coordination site. After 40 generations, even more ligands coordinating *via* nitrile groups are found while the number of ligands containing pyridine derivatives as binding sites decreases. Instead, ligands that coordinate *via* a carbonyl group are preferred. This shows that the GA actually traverses chemical space since the final population contains mainly coordination sites that are not present in the starting population and not just interpolated between chemical structures present in the starting population.

*Schilter et al. (2023)* developed a variational-autoencoder trained on the dataset from *Meyer et al. (2018)* and were able to discover novel catalysts with favourable energy differences by optimizing in a learned latent space. They show that the distribution of coordination sites for the generated molecules follows the distribution of the training data. This indicates, that they find novel ligands by interpolating in the latent space, but do not

-27.46 kcal/mol

-26.82 kcal/mol

-28.31 kcal/mol

-26.25 kcal/mol

**Figure 7** **Best-performing individuals of the GBGA with synthetic accessibility constraint after 40 generations.** The calculated energy difference is shown to the right of the structure.

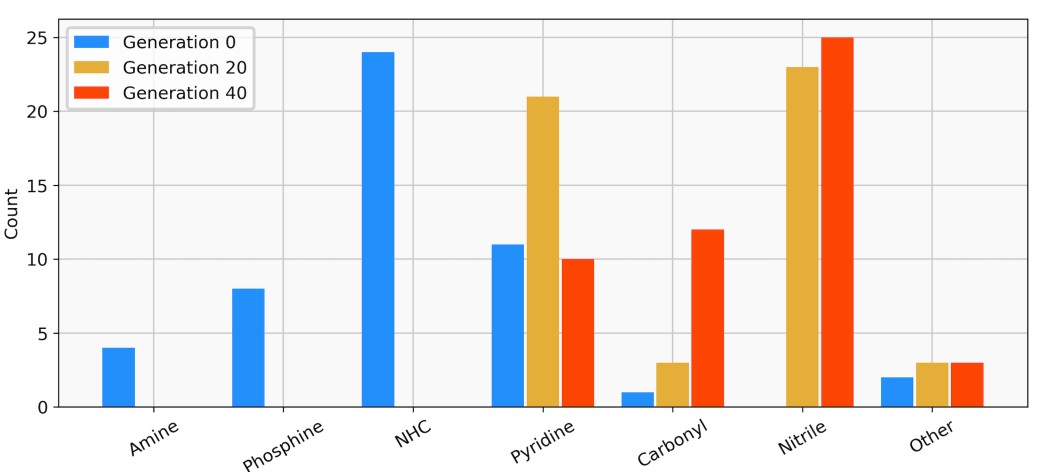

**Figure 8** **Distribution of the functional groups *via* which the ligands bind to the transition metal from a GBGA run with synthetic accessibility constraint for the initial population (blue), after 20 generations (yellow) and for the final population (red).**

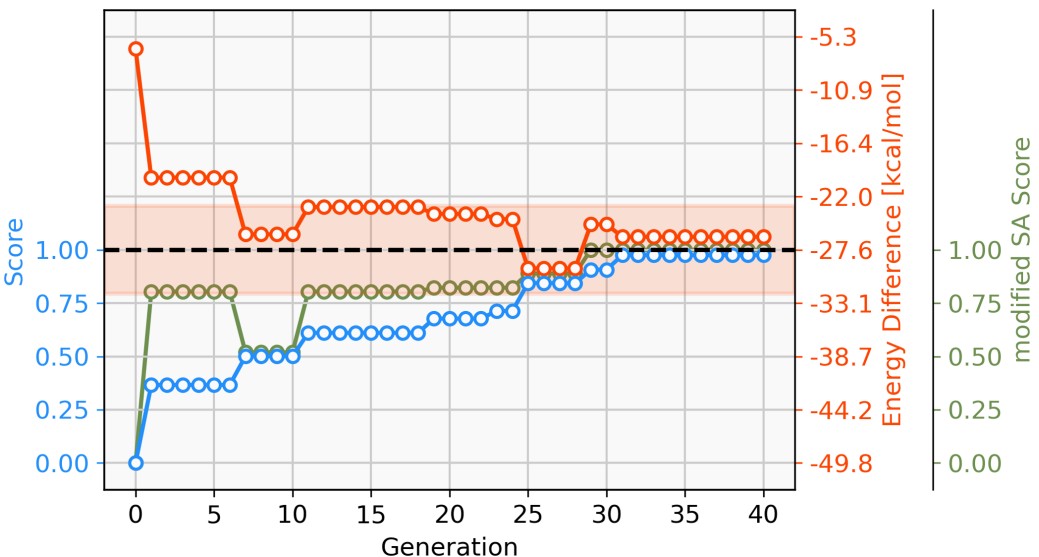

**Figure 9** Evolution of the score (blue), energy difference (red) and modified SA score (green) of the best-performing individual of a GBGA run with copper (Cu) containing catalysts over 40 generations. The target range of the energy difference is shown as a red-shaded area.

discover novel binding motifs in a different area of chemical space than what their training data contains.

## Graph-based GA for copper and silver based catalysts

With a GBGA, it is straightforward to discover novel catalysts utilizing other transition metals than Pd. Here, we show the generation of novel ligands for copper (Cu) and silver (Ag) based catalysts with a favourable thermodynamic profile for the Suzuki reaction. This appears to be a challenging task since *Meyer et al. (2018)* were only able to find 20 and 0 catalysts in the desired energy range *via* screening of 18,062 catalyst candidates, respectively. Furthermore, when calculating the actual energy difference at the B3LYP-D3BJ/def2-TZVP//B3LYP-D3BJ/3-21 level of theory, we could only confirm 6 out of 20 Cu-based catalysts with $\Delta E$ values within $-32.1$ and $-23.0$ kcal mol$^{-1}$. The evolution of the score, calculated energy difference and the modified SA score of the best Cu-based catalysts over 40 generations are shown in Fig. 9. The calculated $\Delta E$ s of the best catalysts in the early generations are considerably higher ($>15$ kcal mol$^{-1}$) than the desired target range which yields low overall scores in the first six generations. After seven generations, a catalyst with a calculated $\Delta E$ in the target range is identified. In the following 33 generations more catalysts within the target range and varying modified SA scores are discovered by tradeoff between the two objectives. Overall, 112 unique catalysts within the target range were discovered by evaluating 1,000 catalysts with DFT, a subset is shown in Fig. 10.

For the generation of Ag-based catalysts, no synthetic accessibility constraint was applied since preliminary experiments proved the generation of catalysts with both high SA scores and $\Delta E$ values in the target range too challenging. We therefore show how the GBGA is used to generate structural motifs that yield catalysts within the desired $\Delta E$ range. Analysis

**Figure 10 Best-performing individuals of the GBGA with synthetic accessibility constraint for Cu-based catalysts after 40 generations.** The calculated energy difference is shown to the right of the structure.

of the generated structures yields insight into electronic and steric effects that would be necessary for Ag-based catalysts.

68 catalysts with ΔE values in the target range could be identified over 40 generations. Figure 11 shows the evolution of the score (blue) and the calculated ΔE (red) of the best catalyst over 40 generations. This task appears to be more challenging than previous ones, since it takes 22 generations without SA constraint until a catalyst with ΔE in the target range is discovered. The final catalyst candidates are all anionic which appears to stabilize the reactant in intermediate **2**. Figure 12 shows the structure of the best-performing catalyst candidate with the interaction between the ligand and reactant in blue. This can be seen as an example of what the structure of the ligand would need to look like in order for the catalyst to fall within the target energy range. The unusual structure with a deprotonated NHC is not expected to be a stable complex and could most likely not be synthesised. Yet, the structure could be useful for further optimization while considering the need for non-covalent interactions between the ligand and the reactant.

## CONCLUSION

In conclusion, the results of our study demonstrate the effectiveness of fragment-based genetic algorithms (FBGAs) and, especially, graph-based genetic algorithms (GBGAs) in the search for novel ligands for catalysts in the Suzuki reaction. The FBGA successfully

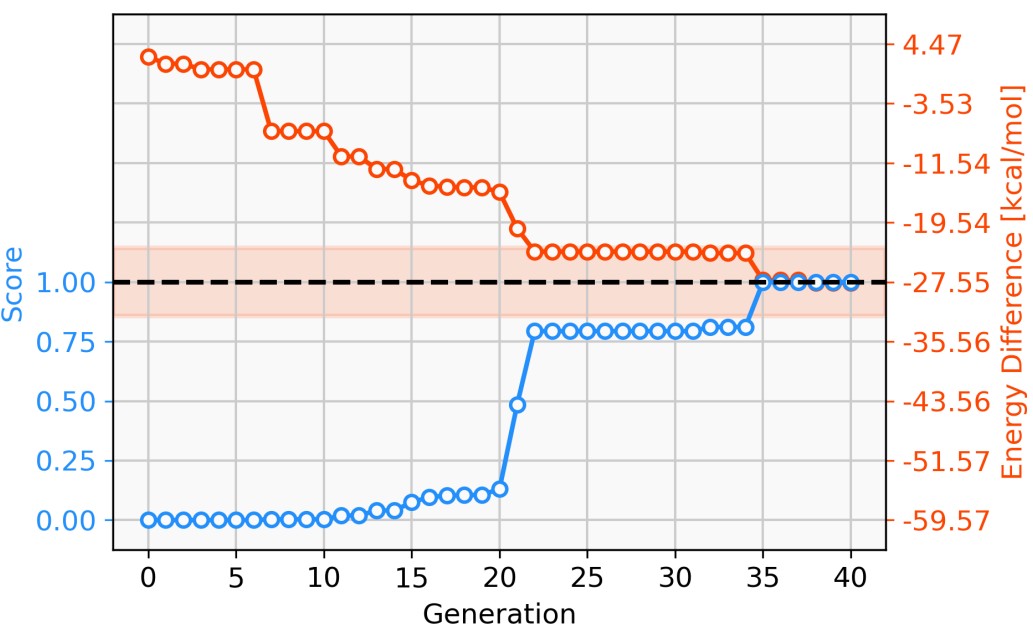

**Figure 11  Evolution of the score (blue) and the energy difference (red) of the best-performing individual of a GBGA run with Ag-containing catalysts over 40 generations.** The target range of the energy difference is shown as a red-shaded area.

**Figure 12  Lewis structure of the best-performing catalyst candidate from GA run containing silver.** The favourable interaction between one ligand and a reactant is shown as a blue dashed line.

evolved a population of molecules over 10 generations, yielding 134 unique catalysts with $\Delta E$ values within the target range. Our study demonstrates that GAs, requiring on the order of 500 evaluations or less, are effective in directly identifying competitive catalysts using DFT without the need for constructing a ML model.

The GBGA, without synthetic accessibility (SA) constraints, explored chemical space by iteratively applying crossover and mutation operations on ligand molecular graphs. The

resulting ligands exhibited a diverse range of coordination sites, emphasizing the capability of GAs to discover novel structures. With SA constraints incorporated in the latter part of the study, the GA navigated towards ligands with improved synthetic accessibility while maintaining a ∆E within the target range. The final population of ligands, identified after 40 generations, showcased diverse binding motifs and confirmed the GA's ability to discover ligands with desirable properties for catalysis.

Furthermore, the application of GBGAs to explore ligands for Cu- and Ag-based catalysts in the Suzuki reaction revealed their potential to generate novel structures for different transition metals. Despite the challenges associated with the limited number of previously identified Cu-based catalysts, the GA successfully discovered 112 unique Cu-based catalysts within the desired ∆E range, demonstrating the versatility of the GA approach for exploring novel catalytic systems.

Generation of Ag-based catalysts with favourable thermodynamic profiles showed that the GBGA can discover structural motifs that yield catalysts within the target ∆E range without regard for stability or synthesizability. These structural motifs can yield insights for molecular discovery of further ligands.

Comparisons with existing ML models highlight the complementarity of GA methods, as GAs traverse chemical space, discovering ligands with coordination sites not present in the initial population. This is in contrast to ML models that interpolate within a learned latent space but may struggle to explore entirely new binding motifs.

As with all generative models, the real-life performance of structures generated by this molecular optimization approach is limited by the applicability of the used scoring function or property prediction model. Here, we optimize the DFT-calculated energy of one specific reaction step in one well-defined catalytic cycle. To assess the real-world performance of a generated catalyst, an extensive computational strategy should be applied to assess the underlying assumptions.

The proposed catalyst structure should first be evaluated through an extensive conformational search covering all possible coordination modes and sites, potentially utilizing CREST for this purpose (*Pracht, Bohle & Grimme, 2020*). Subsequently, the catalyst's stability can be assessed using methods like the (meta-)dynamics approach outlined by *Grimme (2019)*. Also, the dominant oxidation state of the metal center needs to be considered. Next, the applicability of the linear energy scaling relation must be validated for the generated molecules by calculating all reaction intermediates. Additionally, it is crucial to locate the transition states throughout the catalytic cycle and compute their associated activation barriers, thereby avoiding reliance on linear correlations between intermediate energies and actual activation barriers. Finally, the entire reaction network of the reaction system should be investigated to identify any competing side reactions, similar to the work of *Rasmussen, Seumer & Jensen (2023)*.

In summary, our study underscores the potential of genetic algorithms as powerful tools for ligand discovery in catalysis, showcasing their ability to efficiently navigate chemical space, discover novel structures, and generate ligands with desirable properties while minimizing computational costs.
Finally, the employed synthetic accessibility (SA) score in this study is observed to impose penalties on frequently utilized ligands, including phosphines and carbenes, redirecting the exploration towards drug-like chemical space. To achieve a more comprehensive exploration of the relevant chemical space, ongoing research in our group is dedicated to developing synthetic accessibility measures tailored for homogeneous inorganic catalysts.

### Funding
This work was supported by the Novo Nordisk Foundation (NNF20OC0064104). The funders had no role in study design, data collection and analysis, decision to publish, or preparation of the manuscript.

### Grant Disclosures
The following grant information was disclosed by the authors:
Novo Nordisk Foundation: NNF20OC0064104.

### Competing Interests
Jan Jensen is the Editor in Chief for PeerJ Physical Chemistry.

### Author Contributions
- Julius Seumer performed the experiments, analyzed the data, performed the computation work, prepared figures and/or tables, authored or reviewed drafts of the article, and approved the final draft.
- Jan H. Jensen conceived and designed the experiments, analyzed the data, authored or reviewed drafts of the article, and approved the final draft.

### Data Availability
The code and data are available at Github and Zenodo:
- github.com/jensengroup/tmcat-design.
- Julius Seumer. (2024). jensengroup/tmcat-design: v1.0 (v1.0). Zenodo. https://doi.org/10.5281/zenodo.14168671.

### Supplemental Information
Supplemental information for this article can be found online at http://dx.doi.org/10.7717/peerj-pchem.34#supplemental-information.

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
