# Peer review of "Beyond predefined ligand libraries: a genetic algorithm approach for *de novo* discovery of catalysts for the Suzuki coupling reactions"

_PeerJ Physical Chemistry, doi:10.7717/peerj-pchem.34_

## Round 0.1 · original submission · Minor Revisions

Please address the reviewer comments below.

Reviewer 1 ·

Basic reporting

no comment

Experimental design

no comment

Validity of the findings

no comment

Additional comments

This manuscript explored the methods for the de novo design of transition metal catalysts by combining genetic algorithms and first principle calculations. The authors focused on the Suzuki reaction to demonstrate the ability of fragment-based and graph-based genetic algorithms in identifying novel ligands for transition metal catalysis. The main concept of this manuscript is extended from former work in Angewandte Chemie 2023, e202218565 and Chemical science 2019,10,3567-3572. This manuscript is well written, offers detailed raw data and provides the code for transparency and reproducibility of the results, and useful for catalysis community. This paper is suitable to PeerJ Physical Chemistry. I recommend to publish after revision.

There are some comments as follow:

1. There are some spelling errors, such as in line 56, ‘challening’ should be challenging. Figure 1B, ‘optimiation’ should be optimization. The author should check the spelling carefully to match the language standard of PeerJ journal.
2. The reference list should be improved. The authors listed some journals with full names, some are abbreviations, such as line 339/340. Some papers pages are missing such as line 306/312. Please follow the PeerJ journal reference guidelines to revise the format of the references.
3. In figure 1b, the authors declared that ‘all constitutional isomers are generated and their structures optimized with GFN2-xTB. 10 conformers of the lowest energy constitutional isomer are then optimized with GFN2-xTB’.
First, it would be more clear if the authors could briefly introduce how they generate the conformers here, such as ‘all constitutional isomers are generated by embedding using ETKDG in RDKit’.
Second, according to line 93, the authors only generated 10 conformers of intermediate 2. So the 10 lowest energy conformers are the all conformers the authors generated, which is a bit tricky. It would make more sense to describe as ’10 lowest energy conformers’ in the manuscript if the authors generated more than 10 conformers and then choose 10 lowest energy conformers.
Third, why the authors optimized the structures two times with GFN2-xTB? Did the authors use tighter setting for latter geometry optimization or the same settings? It should be mentioned here.
4. Again, for line 77, 25 conformers of each are embedded, how are the conformers embedded?
5. The authors only generated 10 conformers of intermediate 2 using ETKDG as described in line 93. Intermediate 2 is a flexible molecule instead of rigid molecule, therefore, too little conformer sampling could lead to missing the real lowest energy conformer. Could the author check if they generate enough conformers for obtaining lowest energy conformers. It would be interesting to see if generating more conformers such as 50 conformers, still yields the same lowest energy conformer or could result in finding a lower energy conformer for the intermediate.
6. For line 114, the authors defined fitness values, why do the authors choose a selection pressure (SP) of 1.5?
7. For line 118, the starting population is comprised of 25 molecules, it would be intuitive for readers if the authors could add a figure to depict some molecules and ligands. In line 121, what is the full list of available ligands? If it is from reference, adding the reference here could help readers to understand.
8. The GA method could find structures beyond initial populations, it would be nice to show the structure diversity for different generations using such as Chemical space network for better visualization.

Reviewer 2 ·

Basic reporting

1. In Figure 1, it would be better to provide labels for atoms which illustrates intermediate 2 and 1. It seems red ball is a metal atom and the other balls represent ligand molecules but having element label or any other ways that help reader which ball corresponds to which element would be helpful.

2. Regarding synthetic accessibility (SA) score, although it has been introduced in the cited reference, it would be beneficial for the readers to have a bit of explanation how this value is computed for a given ligand molecule.

3. In Figure 2 (and also the other related figures), the best performing score with respect to generation is plotted. I suspect one of catalyst with high score might have been found fortuitously. The excellence of GA could better be represented if (for example) accumulated number of candidates within target range. If this accumulated number constantly increases over generation, this will strengthen the value of this model.

Experimental design

1. The target Delta E value (i.e. -27.55 kcal/mol) which is the energy difference between intermediate 2 and intermediate 1 and R-X has been determined empirically. (Also the sigma value) The authors might want to elaborate which empirical procedure was used to choose those values.

2. The authors explains normalized fitness value (equation 2) which seems to be used for selecting which ligand is selected for mutation. However, its relation to genetic algorithm and the catalyst finding performance is a bit unclear. How does this factor leads to performance of GA? Also the motivation of using this criteria can be also further clarified.

Validity of the findings

no comment

---

## Round 0.2 · accepted · Accept

In light of the revisions, the paper can be accepted for publication in it's current form.

Reviewer 1 ·

Basic reporting

It is very good now

Experimental design

It is very good now

Validity of the findings

It is very good now